# Anti-Inflammatory and Immunomodulatory Effects of 0.1 Sub-Terahertz Irradiation in Collagen-Induced Arthritis Mice

**DOI:** 10.3390/ijms25115963

**Published:** 2024-05-29

**Authors:** Qi Zhang, Sen Shang, Xu Li, Xiaoyun Lu

**Affiliations:** Key Laboratory of Biomedical Information Engineering of the Ministry of Education, School of Life Science and Technology, Xi’an Jiaotong University, Xi’an 710049, China; qqqizhang609408@126.com (Q.Z.); shangsen2106@xjtu.edu.cn (S.S.); lixu15668451108@163.com (X.L.)

**Keywords:** terahertz irradiation, inflammation, immunity, CIA mice, HaCaT

## Abstract

The primary emphasis of photoimmunology is the impact of nonionizing radiation on the immune system. With the development of terahertz (THz) and sub-terahertz (sub-THz) technology, the biological effects of this emerging nonionizing radiation, particularly its influence on immune function, remain insufficiently explored but are progressively attracting attention. Here, we demonstrated that 0.1 sub-THz radiation can modulate the immune system and alleviate symptoms of arthritis in collagen-induced arthritis (CIA) mice through a nonthermal manner. The application of 0.1 sub-THz irradiation led to a decrease in proinflammatory factors within the joints and serum, reducing the levels of blood immune cells and the quantity of splenic CD4^+^ T cells. Notably, 0.1 sub-THz irradiation restored depleted Treg cells in CIA mice and re-established the Th17/Treg equilibrium. These findings suggested that sub-THz irradiation plays a crucial role in systemic immunoregulation. Further exploration of its immune modulation mechanisms revealed the anti-inflammatory properties of 0.1 sub-THz on LPS-stimulated skin keratinocytes. Through the reduction in NF-κB signaling and NLRP3 inflammasome activation, 0.1 sub-THz irradiation effectively decreased the production of inflammatory factors and immune-active substances, including IL-1β and PGE_2_, in HaCaT cells. Consequently, 0.1 sub-THz irradiation mitigated the inflammatory response and contributed to the maintenance of immune tolerance in CIA mice. This research provided significant new evidence supporting the systemic impacts of 0.1 sub-THz radiation, particularly on the immune system. It also enhanced the field of photoimmunology and offered valuable insights into the potential biomedical applications of 0.1 sub-THz radiation for treating autoimmune diseases.

## 1. Introduction

The primary focus of photoimmunology research is the impact of nonionizing radiation on the immune system. Originating from the 1970s and 1980s, early research in this field mainly concentrated on UV-induced skin cancer and immune tolerance. Yet, current research on the immune effects and mechanisms of other forms of nonionizing radiation remains very limited. Being the most recently developed nonionizing radiation that lies between microwave and infrared frequencies, the biological effects of sub-terahertz (sub-THz) and THz radiation are of particular interest and are increasingly receiving attention in the scientific community. The inquiry into the potential impact of sub-THz and THz radiation on the body’s immune system remains unresolved. Previous studies have provided preliminary but interesting insights, including changes in the expression of immune-related genes, the initiation of inflammatory responses, and the hindrance of the wound healing process. For instance, the up-regulation of interleukins such as IL-26 and IL-5 was observed in Jurkat cells exposed to 2.52 THz radiation [1]. Additionally, exposure to frequencies of 1.4, 2.52, and 3.11 THz altered the functional pathways involved in cellular immune responses in human keratinocytes [2]. Analysis of artificial human skin tissues revealed significant dysregulation in the expressions of nearly half of the epidermal differentiation complex members, involved in skin immune and inflammatory responses, and the cytokine–cytokine receptor interaction pathway following exposure to intense broadband THz pulses [3]. Moreover, in vivo experiments demonstrated that pulsed THz waves (2.7 THz, with an average power of 260 mW/cm^2^) led to a substantial recruitment of infiltrating neutrophils in the mouse’s ear skin [4], and repeated exposure to fs-THz radiation (2.5 THz, 0.32 mW/cm^2^) delayed the wound healing process in the skin of THz-irradiated mice [5]. Although the precise genes or pathways affected by various THz radiation were not consistent in the aforementioned studies, it still naturally led to speculation that sub-THz and THz radiation may trigger cellular responses and the production of certain immune signals. This, in turn, could potentially induce crosstalk between the impacted cells and nearby immune cells and possibly the entire immune system.

In our prior research, we observed a notable decrease in multiple serum proinflammatory cytokines and chemokines, including interleukin-6 (IL-6) and CXCL1, in mice subjected to abdominal 3.1 THz irradiation (unpublished data). It implied that that the localized effects of THz irradiation may also contribute to modulating systemic inflammation and influencing the immune response. Inspired by these observations and related reports, we conducted an investigation into the impact of sub-THz irradiation on inflammatory manifestations in the collagen-induced arthritis (CIA) mice, which is indicative of the autoimmune disease, rheumatoid arthritis (RA). RA is distinguished by an excessive systemic inflammatory reaction, characterized by the accumulation of inflammatory cells in the joints, persistent synovitis, and irreversible damage to cartilage and bone [6,7]. The overproduction of proinflammatory cytokines and chemokines, including IL-6, IL-17, and tumor necrosis factor alpha (TNF-α), accelerates various inflammatory processes in the development of RA [8]. However, this overreaction inhibits the differentiation of regulatory T cells (Tregs), resulting in compromised immune tolerance and heightened immune responses in RA [9].

In the present study, the ankles of CIA mice were exposed to a 0.1 sub-THz continuous-wave irradiation for a duration of 30 min daily over a period of two consecutive weeks. Our findings indicated that, following the 0.1 sub-THz irradiation, there was a significant reduction in both ankle swelling and inflammation in the CIA mice. Moreover, beyond the localized alleviation of ankle inflammation, a reduction in systemic inflammatory markers, a decrease in the splenic CD4^+^ T cell population, and a concurrent recovery of splenic Treg cells were also observed. Analysis of blood leukocyte mRNA sequencing data revealed that the top 10 biological processes enriched from differentially expressed genes (DEGs) were all associated with immune response. These outcomes demonstrated that 0.1 sub-THz radiation exerted a protective effect on mice afflicted with RA and immune hyperactivation, exhibiting both anti-inflammatory and immunomodulatory capabilities. The findings of this study have not only expanded the current understanding of sub-THz bioeffects but also shed new light on the interplay between sub-THz waves and the immune system. Of particular note is the potential of sub-THz technology in the management of RA, underscoring its auspicious prospects in the biomedical field. Collectively, these discoveries undoubtedly augment the body of knowledge within photoimmunology, thereby broadening the scope of the field to encompass the sub-THz spectrum.

## 2. Results

### 2.1. Irradiation of 0.1 Sub-THz Attenuated Joint Swelling and Cartilage Damage in CIA Mice

After 14 consecutive days of 0.1 sub-THz irradiation for 30 min per day on the swollen ankle joint of CIA mice, the alteration in swelling degree of the ankle joint was the primary indicator of the anti-inflammatory effects of sub-THz against RA. As shown in Figure 1, the CIA mice exhibited severe joint swelling in comparison to the control mice. Meanwhile, 14 days of 0.1 sub-THz irradiation significantly alleviated the arthritis swelling of CIA mice, as evidenced by the decreased average paw thickness and arthritic score (Figure 1A,B). There was also a notable alleviation in joint swelling and redness (Figure 1D). In addition, the body weight of CIA mice, which was monitored every three days, also showed a significant recovery after 0.1 sub-THz irradiation compared to the CIA group (Figure 1C). The alterations observed were attributed to a nonthermal effect of 0.1 sub-THz radiation, as the maximum temperature difference during the exposure was less than 1 °C (Appendix A). In addition, the expression of heat shock proteins HSP70 and HSP90 analyzed by IHC staining was not changed in the area irradiated by 0.1 sub-THz (Appendix A). Therefore, it showed that a two-week exposure to 0.1 sub-THz irradiation significantly attenuated the symptoms of rheumatoid arthritis possibly via a nonthermal mechanism.

Additionally, the effects of 0.1 sub-THz irradiation on the ankle joints were further analyzed and evaluated histopathologically. The CIA mice showed severe joint infiltration, heightened synovitis, and pannus formation upon tissue slide examination with H&E staining. Additionally, noticeable cartilage damage could be detected in the joints of CIA mice via Toluidine Blue O staining (Figure 1D). Nevertheless, in the CIA+THz group, the inflammatory infiltration and cartilage destruction were significantly ameliorated by sub-THz irradiation, ultimately leading to a reduction in synovitis and joint damage (Figure 1D).

### 2.2. Irradiation of 0.1 Sub-THz Reduced Proinflammatory Cytokine Levels in the Ankle Joint

Meanwhile, the leukocyte marker CD45 and the antigen-presenting cell (APC) marker MHC II were detected by IF and IHC staining. It could be observed that 0.1 sub-THz irradiation resulted in a decrease in the number of infiltrated leukocytes and the expression of MHC II by APCs in the ankle joint of CIA mice (Figure 2A–C). To further confirm the anti-inflammatory effect of 0.1 sub-THz on the ankle joint arthritis of CIA mice, the levels of TNF-α, IL-6, and IL-17, which extensively contribute to the progression and severity of RA, were analyzed. As shown in Figure 2, all of the TNF-α, IL-6, and IL-17 were remarkably reduced in the joints of the CIA mice after sub-THz irradiation (Figure 2D–G). In addition, the protein levels of matrix metalloproteinases MMP3 and MMP9, as well as receptor activator for NF-kB ligand (RANKL), which act downstream of IL-6 and contribute to joint damage and expedite inflammation, were significantly decreased in joint tissue after sub-THz irradiation (Appendix A). In contrast, bone morphogenetic protein 2 (BMP2), a vital marker of bone remodeling involved in cartilage repair, was slightly elevated in the CIA group but significantly increased after sub-THz exposure (Appendix A). These findings indicated that THz irradiation can decrease the levels of proinflammatory factors and enhance the expression of proteins related to bone remodeling, thereby alleviating arthritis damage and aiding in tissue repair to a certain extent.

### 2.3. Irradiation of 0.1 Sub-THz Attenuated the Leukocytes Enhancement and Altered the Gene Expression of Blood Leukocytes in CIA Mice

RA, as a typical autoimmune disease, is characterized by not only joint-related symptoms but also a heightened systemic inflammatory response. To determine whether a local sub-THz treatment could potentially have systemic regulatory effects, the quantities of white blood cells, neutrophils, lymphocytes, and monocytes in each group were analyzed using a semi-automatic biochemical analyzer. As shown in Figure 3, the elevated counts of lymphocytes, neutrophils, monocytes, and total white blood cells in CIA mice indicated a hyperactive immune response and a heightened inflammation in vivo (Figure 3A–D). Meanwhile, in the CIA+THz group, there was a clear reduction in systemic inflammation, as indicated by the decreased enhancement of various types of leukocytes (Figure 3A–D).

To gain a better understanding of how white blood cells respond to sub-THz irradiation, sequencing and transcriptome analysis of circulating hemocyte mRNA was conducted. A total of 2287 DEGs were identified among the Ctrl, CIA, and CIA+THz groups, based on a selection criterion of *p* < 0.05 (Appendix A), and were then subjected to GO enrichment and KEGG analysis. Figure 3E displays the top 10 enriched GO terms in each of the biological processes (BP), cellular components (CC), and molecular functions (MF) categories among the three groups. It could be noticed that most of the enriched biological processes were associated with the immune response, such as leukocyte differentiation and adhesion, lymphocyte differentiation, and T cell activation. In addition, many other immune-related biological processes, including “positive regulation of cytokine production”, “T-cell differentiation”, and “αβ T-cell activation”, were also significantly enriched (listed in Appendix A). Meanwhile, the KEGG analysis of the DEGs showed enrichment in the “Chemokine Signaling Pathway”, “Th17 Cell Differentiation”, and “Rheumatoid Arthritis” (Figure 3F and Appendix A).

To further refine our findings, DEGs between the CIA and CIA+THz groups (identified using a *p*-value < 0.05 and a |log2 Fold Change| > 0.67, as listed in Appendix A) were further subjected to GO enrichment analysis (Figure 3G and Appendix A). The terms of “negative regulation of T cell-mediated immunity”, “regulation of T cell activation”, and “lymphocyte proliferation” were clearly among the top significant BP categories. In addition, “cell-cell adhesion” and “interleukin-10 production” were also enriched. Correspondingly, significant changes in molecule binding, such as chemokine receptor binding, cytokine binding, and integrin binding, were observed among the top enriched MF categories (Figure 3G). In the CC category, many DEGs were targeted to the Golgi membrane and plasma membrane. Furthermore, the essential DEGs involved in the top 20 significant BP categories were displayed on the Sankey dot plot (Figure 3H). Among them, a wide range of cytokines, chemokines, and regulatory factors, including CD28, CD40lg, IL-6, CXCL12, IL-13, RORc, and Btla, were observed to be involved. The variations in both leukocyte levels and the gene expression status of these cells implied that 0.1 sub-THz irradiation induced a systemic alteration of the immune status.

### 2.4. Irradiation of 0.1 Sub-THz Reduced the Levels of Proinflammatory Cytokines in the Serum

As an inflammatory disease, the content of characteristic cytokines in RA represents the level of inflammation in vivo. Therefore, the levels of IL-17, TNF-α, IL-6, and IL-1β in the serum, which generally reflect the degree of systemic inflammation in RA, were then measured using ELISA. These cytokine levels are indicative of the overall level of inflammation in the body. As expected, 0.1 sub-THz irradiation reduced the excessive accumulation of IL-17, IL-6, and TNF-α in the serum of CIA mice (Figure 4A–C). A similar change was also observed for IL-1β (Figure 4D). In addition, several other cytokines and chemokines associated with autoimmune diseases were significantly altered. It was found that the serum levels of chemokine ligand CXCL13, CXCL12, and CXCL1, interleukin family factors IL-1ra, IL-16, and IL-1α, as well as M-CSF, TIMP-1, and TREM-1 were all significantly increased in the CIA group but decreased in the CIA+THz group (Figure 4E). The serum level of the complement component C5a was significantly decreased in the CIA group; however, it reverted back to a normal level in the CIA+THz group (Figure 4E). These findings intensified speculation that 0.1 sub-THz irradiation could not only relieve arthritis symptoms but also have a systemic anti-inflammatory effect.

### 2.5. Radiation of 0.1 Sub-THz Reduced Immune Activation in the Spleen of CIA Mice

The spleen is a vital organ of the immune system responsible for maintaining immune homeostasis in the body. To further confirm the systemic anti-inflammatory effect of 0.1 sub-THz irradiation, the weight of the spleen was measured and spleen tissue sections were analyzed. It was found that 0.1 sub-THz irradiation greatly attenuated spleen enlargement in CIA mice and significantly reduced the spleen organ coefficient (Figure 5A,B). Histological analysis also showed that the control group exhibited a normal tissue structure, including clear white pulps, red pulps, and marginal zones with few macrophages present (Figure 5C). In contrast, the CIA group exhibited a massive accumulation of macrophages, which was also alleviated in the CIA+THz group (Figure 5C).

The pathogenesis and progression of RA is characterized by hyperactive CD4^+^ T cells, which are essential to the autoimmune response. Combined with the consideration of the GO analysis results, which implied that T cell proliferation and activation have been altered, the percentages of CD4^+^ and CD8^+^ T cells in the spleen were identified by flow cytometry analysis. As expected, the CIA mice displayed a greater proportion of CD4^+^ T cells without a remarkable increase in CD8^+^ T cells (Figure 5D–F). Irradiation of 0.1 sub-THz caused a considerable decrease in the number of CD4^+^ T cells, without any observable effect on the percentage of CD8^+^ T cells (Figure 5D–F). Consequently, the CD4^+^/CD8^+^ ratio significantly decreased after sub-THz exposure (Figure 5D,G). Moreover, analysis of activated and naïve T cells in the spleen showed that the number of activated T cells (CD69^+^) was significantly higher in the CIA group than in the control group, while 0.1 sub-THz irradiation reduced the level of activated T cells (Appendix A). Meanwhile, the amount of naïve T cells (CD62L^+^) was decreased in the CIA group, and 0.1 sub-THz irradiation had no impact on it (Appendix A). These results suggested that sub-THz irradiation could suppress immune activation in the spleen, potentially by hindering the activation and expansion of CD4^+^ T cells.

In addition, RA is associated with a reduction in the percentage of regulatory T cells (Tregs), which are essential for maintaining immune homeostasis and avoiding excessive immune reactions. Further analysis of the percentage of Treg cells, identified as CD4^+^CD25^+^FOXP3^+^, through flow cytometry demonstrated a partial effect of 0.1 sub-THz irradiation in restoring the number of Treg cells in the spleens of the CIA+THz group compared to that of the CIA group (Figure 5H,I). The spleen’s IF detection of FOXP3^+^ signals also confirmed an increase in the number of Treg cells (Appendix A). Meanwhile, since the balance between Treg cells and Th17 cells is crucial for the pathogenesis of RA, the Th17 cells (IL-17^+^) in the spleen were analyzed using IF and flow cytometry. Exposure of 0.1 sub-THz led to a decrease in the percentage of Th17 cells in the spleen of CIA+THz mice (Figure 5J,K, Appendix A). It also coincided with the observation that 0.1 sub-THz exposure resulted in a decreased IL-17 level in the serum and reduced expression of RORc, which codes for a Th17-specific transcription factor, in CIA+THz mice. These results implied that 0.1 sub-THz irradiation shifted the balance between Treg cells and Th17 cells towards a more immune-tolerant state. Taken together, these data suggested that sub-THz irradiation had an immunosuppressive effect, inhibiting the activation of CD4^+^ T cells and increasing the proportion of Treg cells in CIA mice. This facilitated the restoration of self-tolerance and ultimately reduced the inflammatory response in CIA mice.

### 2.6. Irradiation of 0.1 Sub-THz Elevated the LPS-Stimulated Keratocytes Inflammatory Responses

The above results clearly demonstrated the immunological regulatory effects of 0.1 sub-THz irradiation. Taking into account the capacity of sub-THz radiation to penetrate, we hypothesized that the impact of sub-THz on immune status is an indirect outcome. It is more like a secondary effect, mediated by the response of the irradiated local skin tissue. To test our hypothesis, we employed an immortalized keratinocyte cell line, HaCaT cells, as a model to evaluate the effects of 0.1 sub-THz irradiation on LPS-induced keratinocyte inflammation. LPS can activate the NLRP3 inflammasome as well as the NF-κB signaling pathway by binding to TLR4, resulting in increased production and maturation of caspase-1 and IL-1b in HaCat cells. It was found that, under the inflammatory condition caused by LPS, 0.1 sub-THz irradiation also had a considerable effect on alleviating the LPS-induced inflammatory responses. This was indicated by the decreased expression and subsequent maturation of caspase-1 and IL-1β in HacaT cells (Figure 6A–E). At the same time, the activation of NF-kB, as reflected by the pp65 level, was also reduced under 0.1 sub-THz exposure (Figure 6F,G). The observed diminished transcription levels of IL-1β, IL-6, and IL-8 further validated the alleviation of LPS-induced inflammatory response inflicted by sub-THz irradiation (Figure 6H). Moreover, the overexpression of COX-2, which is induced by the activation of the NLRP3 inflammasome and NF-kB signaling, and the consequent production of PGE_2_ were also decreased after 0.1 sub-THz exposure (Figure 6I–K). All these findings corroborated that sub-THz irradiation attenuated the production of proinflammatory cytokines and immune-active molecule such as PGE_2_ under inflammatory stimulation in HaCaT cells. It could be speculated that sub-THz irradiation could attenuate the activation of local and systemic immune cells in the body by reducing the production of proinflammatory factors and immune-active mediators by keratinocyte cells. This could subsequently contribute to the systemic immune regulatory effects.

## 3. Discussion

As the most recently developed frequency band in the electromagnetic spectrum, sub-THz and THz radiation has driven heightened curiosity in the biological effects and the biomedical applications; yet, there are few studies that have been conducted to assess the efficacy of sub-THz intervention for diseases. Although several previous studies have already shown that THz waves can influence the expression of genes associated with the cellular immune response [10,11], there is still a lack of knowledge on how sub-THz and THz affects the immune system, particularly under certain inflammatory conditions. This study presented solid proof that 0.1 sub-THz exposure had a positive influence on rheumatoid arthritis in CIA mice, which was used as a mouse model of human RA. Results showed that a nonthermal, local, and systemic impact was observed when CIA mice were exposed to continuous-wave THz irradiation at a frequency of 0.1 sub-THz, reducing the severity of rheumatoid arthritis in CIA mice. Irradiation of 0.1 sub-THz not only alleviated the joint swelling and diminished the expression of key inflammatory cytokines in the joints of CIA mice but also decreased the systemic inflammatory response, with a decrease in the number of leukocytes in the blood and the levels of inflammatory cytokines and chemokines in the serum. Furthermore, the decrease in the proportion of Th17 cells and the recovery of the proportion of Treg cells in CIA mice suggested that 0.1 sub-THz irradiation helped to restore immune balance in CIA mice. These results demonstrated that the anti-inflammatory activity and the immunomodulatory function of 0.1 sub-THz irradiation could be a potential and promising intervention for rheumatoid arthritis.

Excess expression and production of the representative inflammatory cytokines, such as TNFa, IL-6, and IL-17, intensify the inflammatory response and exacerbate joint cartilage destruction in the pathogenesis of RA [12]. On the one hand, cytokines can cause the infiltration and sustained activation of inflammatory cells; on the other hand, they can also result in the production of enzymes that damage bone and cartilage. Our study discovered that 0.1 sub-THz irradiation had a considerable impact on the reduction in proinflammatory cytokines like TNFα and IL-6 in the joint of CIA mice. This was co-ordinated with the decreased expression of MMPs and RANKL, as well as the reduced infiltration of leukocytes in the joint. The decreased production of cytokines, the reduced infiltration of leukocytes, and the alleviated tissue damage interact with one another. We still lack a comprehension of the precise mechanism by which 0.1 sub-THz irradiation induces these alterations. The outcomes of in vitro cell experiments give us a glimpse into what might happen under sub-THz exposure. The inflammatory response of HaCat cells following LPS stimulation was attenuated when 0.1 sub-THz irradiation was applied. The activation of NLRP3 inflammasome and NF-κB signaling pathway, which are triggered by LPS, were mitigated, leading to the decreased production of IL-1b, IL-6, and IL-8. Meanwhile, COX-2 and PGE_2_ were also suppressed by 0.1 sub-THz irradiation. Although the deeper mechanism behind the anti-inflammatory effect of sub-THz against LPS stimulation has yet to be discovered, it presents a potential avenue for further investigation into the in vivo phenomenon observed in the CIA+THz mice.

Being an autoimmune disorder, rheumatoid arthritis is associated with a systemic immune reaction. We discovered that 0.1 sub-THz exposure also had a considerable impact on reversing the systemic inflammation status of CIA mice. In addition to the decreased number of lymphocytes, monocytes, and neutrophils in the blood, the number and the ratio of CD4^+^ T cells in the spleen were also significantly decreased. The GO analysis of DEGs from the transcriptome data provided supporting evidence for the alteration in leukocyte proliferation and T cell activation after 0.1 sub-THz exposure. Meanwhile, the splenic CD69^+^ activated T cells dramatically decreased and reverted to the same level as the mice in the control group. Actually, we also found that the expression of MHC II in the joint of CIA mice was decreased after 0.1 sub-THz exposure (Appendix A), suggesting that the activity of antigen-presenting cells (APCs) was reduced, thus leading to a diminished T cell activation stimulus. The decline in APCs activity was not unexpected. As shown in our HaCat-cell-based experiment, the expression of COX-2 and the production of proinflammatory mediator PGE_2_ were reduced due to 0.1 sub-THz treatment. The reduction in PGE_2_ production, as well as the decreased proinflammatory cytokines, can weaken the activation of immune cells such as dendritic cells, macrophages, and lymphocytes [13]. Taken together, the decreased proinflammatory factors and the less activated APCs caused by 0.1 sub-THz irradiation provide an explanation for the decrease in splenic CD4+ T cells and CD69^+^ activated T cells.

Furthermore, Th17 and Treg, the subsets of CD4^+^ T helper cells, and particularly their balance play a pivotal role in rheumatoid arthritis. Th17 cells are the primary source of IL-17, which exerts powerful proinflammatory and joint-destructive activities. Treg cells have been shown to suppress the proliferation of autologous CD4^+^ T cells, thus preventing autoimmune overactivation and maintaining immune homeostasis [14,15]. Our results revealed that the increased IL-17 in both the joint and serum of CIA mice was obviously reversed by 0.1 sub-THz irradiation, suggesting that the activity of Th17 cells was modified due to sub-THz irradiation. Analysis of the blood leukocytes transcriptome revealed that several biological processes associated with the immune response, differentiation, and lineage commitment of Th17 cells, as well as the differentiation of Treg cells, were enriched among the DEGs between the three groups. Results from flow cytometry analysis ultimately provided firm evidence indicating the reduction in Th17 cells and the recovery of Treg cells, which corroborated all the indications suggested previously and suggested a gradual recovery of immune homeostasis after sub-THz irradiation. The shift in the proportion of Th17 and Treg cells is undoubtedly beneficial in ameliorating rheumatoid arthritis, but further exploration is still necessary to uncover the deeper mechanisms. Regarding the current results, the alteration in some proinflammatory cytokines such as IL-6, TNF-α, and IL-1b might provide at least part of the possible reasons for the shift of the Th17/Treg balance, since all these vital cytokines promote the generation of Th17 cells from naïve T cells, while suppressing the differentiation and functions of Treg cells [16,17]. For sure, the intricate network regulating the Th17/Treg balance involves much more elements; yet, further research is needed to explore the associated mechanisms in greater detail. Nevertheless, the above results highly demonstrated the immunological regulatory effects of 0.1 sub-THz irradiation.

To summarize, these data provide important insights into the potential therapeutic benefits of 0.1 sub-THz irradiation on CIA mice. However, due to the restricted penetration depth of THz waves, the intricate mechanism and potential mediators responsible for transmitting immunoregulatory signals remain to be fully elucidated. Additionally, it is imperative for future research to address the scalability of THz radiation’s anti-inflammatory properties to larger animals and humans, with a more precise evaluation of the dosage–effect relationship. This will ultimately pave the way for clinical translation.

## 4. Conclusions

In conclusion, our findings in this study demonstrated the anti-inflammatory and immunological regulatory effects of 0.1 sub-THz irradiation in autoimmune rheumatoid arthritis mice. Irradiation of 0.1 sub-THz alleviated arthritis symptoms and reduced the expression of several pivotal proinflammatory cytokines, including IL-6, TNF-α, IL-1b, and IL-17 in CIA mice. Excitingly, local effects of sub-THz irradiation could be amplified and transmitted, leading to a reduced systemic inflammatory response and improved immune homeostasis in CIA mice. Exposure of 0.1 sub-THz attenuated the activation of CD4^+^ T cells, reducing the number and activity of Th17 cells and increasing the proportion of Treg cells, indicating that 0.1 sub-THz irradiation had an immunosuppressive effect in CIA mice. These data affirmed the biological effects of sub-THz radiation, giving rise to fresh evidence for its potential biomedical applications. Nevertheless, it also presents more challenges and possibilities to investigate the underlying mechanisms.

## 5. Materials and Methods

### 5.1. Ethical Approval

Animal experiments were in accordance with the guidelines of the National Institutes of Health for the care and use of laboratory animals. The use of animals was approved by the Ethics Committee of Xi’an Jiaotong University (Xi’an, China) (No.xjtu2020.7). All efforts were made to minimize the number of animals and the intensities of noxious stimuli used to show consistent treatment effects.

### 5.2. Animals and Reagents

A total of 60 6-week-old male DBA/1 mice were acquired from the GemPharmatech Co., Ltd. (Nanjing, China). A suitable temperature, adequate sterile food and water, and an SPF environment with reasonable lighting were maintained throughout the entire experiment. Chick type II collagen, Complete Freund’s Adjuvant (CFA), and Incomplete Freund’s Adjuvant (IFA) were bought from Chondrex, Inc. (Woodinville, WA, USA). Acetic acid was bought from Aladdin (Shanghai, China).

### 5.3. Collagen-Induced Arthritis (CIA) Model Establishment and Experimental Design

A total of 60 7-week-old male DBA/1 mice were divided into three groups randomly, with 20 in each group, as follows: the control group (Ctrl), the CIA model group (CIA), and the CIA with THz irradiation group (CIA+THz). The model establishment method is refers to the reagent instructions and a previous reference [18]. Briefly, equal volumes of CFA and Chick type II collagen made a usable emulsion drop. A total of 0.1 mL of injectant was executed through multiple injection points at the tail root for initial immunization, denoted as Day0. A booster injection was administered on Day 21 with the same emulsion of collagen and IFA. About 10 days after the booster immunization, mice gradually began to suffer from joint swelling, indicating the success of the model.

### 5.4. Irradiation Treatment of 0.1 Sub-THz and Real-Time Temperature Monitoring

The center frequency of the THz source (Terasense, San Jose, CA, USA) is 0.1THz, with an average power of 33 mW/cm^2^. The swollen ankle joints of mice were subjected to 14 days of THz irradiation for 30 min per day. During the irradiation process, the anesthesia state of the mouse was maintained by a R500 small animal respiratory anesthesia machine (RDW, Shenzhen, China) filled with isoflurane (RDW, Shenzhen, China). Real-time temperature at the irradiated site was monitored and recorded by the FOTRIC 323PRO (FOTRIC, Shanghai, China) infrared thermal imager.

### 5.5. Arthritic Score, Paw Thickness, Body Weight, and Spleen Organ Index Assessment

RA severity was evaluated by an arthritic score and paw thickness as described previously [18,19]. Briefly, animals were evaluated every 3 days from the booster injection immunization day for signs of arthritis. Each ankle joint was evaluated by a blinded observer and scored individually on a scale of 0–4, with 0 indicating no evidence, 1 indicating erythema, 2 indicating erythema and mild swelling, 3 indicating erythema and moderate swelling, and 4 indicating the most severe swelling and inflammation. All parameters were added per mouse, resulting in a severity score between 0 and 16. Similarly, measurement of the paw thickness with a caliper was conducted every 3 days to obtain objective data on the severity of arthritis manifestation, referred to previously [19]. The body weights of mice were recorded every 3 days after booster immunization. After execution, the mice spleens were carefully peeled off and obtained intact for weighing and further experiments. The splenic organ index was calculated by the ratio of the spleen weight to the body weight.

### 5.6. Histopathology, Immunohistochemistry, and Immunofluorescence Analysis

Histological assessment was used to evaluate changes in the tissue structure. Briefly, the spleens and the irradiated unilateral ankle joints were carefully removed and ensured to be intact. After 4% paraformaldehyde fixation (ankle tissue with pretreated decalcification), the spleen and the ankle joints were cut into thin sections and stained with hematoxylin and eosin. Toluidine Blue O staining was used to evaluate cartilage damage as described previously [20]. Significance areas were exhibited for suitable magnification.

Immunohistochemistry (IHC) and immunofluorescence (IF) analyses were used to evaluate the expression levels of critical cytokines in the ankle joint and spleen, as described previously [21,22]. Simply, after paraffin dehydration, antigen repair, and background sealing, paraffin-embedded tissue sections were incubated with specific polyclonal primary antibodies HSP70, HSP90, IL-17, Foxp3, TNF-α, IL-6, RANKL, BMP2, MMP3, MMP9, CD62L, CD69, CD45, and MHC-II overnight at 4 °C, with secondary antibodies HRP-labeled Goat Anti-Mouse IgG (H+L) (1:2000, CST, USA) for IHC staining or Goat Anti-Rabbit IgG H&L (Alexa Fluor^®^ 488) (1:800, Abcam, Waltham, MA, USA) for IF staining, respectively. Detailed information on antibodies can be found in Appendix A. Representative areas from the visual images of samples were captured and exhibited at a suitable magnification.

### 5.7. Mouse Hemocyte Composition Analysis and Serum Cytokines Assay

Mouse hemocytes were obtained from the eyeball vein. The portions of the blood cells were detected in terms of the number of leukocytes, lymphocytes, monocytes, and neutrophils by a semi-automatic blood biochemical analyzer (Mindray, Shenzhen, China) as described previously [23]. Serum precipitation from another portion of blood was used to detect the content of IL-6, TNF-α, IL-1β, and IL-17 in mouse serum by commercial ELISA kits (R&D Systems, USA) according to the manufacturer’s instructions. Key cytokines and chemokines of autoimmune diseases in the serum were tested using a Mouse Cytokine Array Panel A Array Kit (R&D Systems, Minneapolis, MN, USA) according to the manufacturer’s instructions.

### 5.8. Mice Splenic Lymphocytes Separation and Flow Cytometry Analysis

Mice splenic lymphocytes were separated with a commercial kit (DaYou, Beijing, China) according to the manufacturer’s instructions. Anti-mouse CD4-FITC (Biolegend, San Diego, CA, USA) and anti-mouse CD8-PE (Biolegend, USA) surface staining of PBMCs were performed to evaluate the proportions of CD4+ and CD8+ T cells. For Treg cell identification, lymphocytes were first stained with a surface anti-mouse-CD4-FITC (Biolegend, USA) and an anti-mouse-CD25-APC (Biolegend, USA) according to the instructions. Then, cells were fixed and permeabilized with the True-Nuclear™ Transcription Factor Buffer Set (Biolegend, USA) and intracellularly stained with an anti-mouse-Foxp3-PE (Biolegend, USA). For Th17 cell identification, lymphocytes were first stained with a surface anti-mouse-CD4-FITC (Invitrogin, Waltham, MA, USA) and then fixed and permeabilized with the Fixation/Permeabilization Buffer Set (Invitrogin, USA) and intracellularly stained with an anti-mouse-IL17-PE (Invitrogin, USA). Samples were analyzed with a FACSVerse flow cytometer (USA), and the data were analyzed with FlowJo software Version 10 (Tree Star, Woodburn, OR, USA).

### 5.9. Murine Hemocytes mRNA Sequencing and Bioinformatics Analysis

Total blood RNA samples were extracted using the TruSeq RNA Sample Preparation Kit V2 (Illumina, San Diego, CA, USA) in strict accordance with the manufacturer’s instructions. Total RNA quality, including concentration and purity, was checked to meet the standards. Further purification was carried out using Agencourt AMPure XP-PCR Purification Beads (Beckman Coulter, Brea, CA, USA). A library was constructed on the Illumina HiSeq 2500 (Illumina Inc., San Diego, CA) platform and RNA-sequencing procedures were performed in a 2 × 150 bp paired-end sequencing mode and obtained FastQ data [24]. Raw data processing was analyzed referring to previous references [25,26]. Differentially expressed genes (DEGs) among the Ctrl, CIA, and CIA+THz groups were defined through ANOVA analysis with a *p*-value < 0.05. DEGs between each two experimental groups were filtered with a *p*-value < 0.05 and a |log2 (fold change)| > 0.67. Subsequently, Gene Ontology (GO) and Kyoto Encyclopedia of Genes and Genomes (KEGG) analyses were performed to describe the functions and processes of the above DEGs by using the R software version 4.0.3.ggplot2 package. The results of the above analysis were visualized through the online website (http://www.bioinformatics.com.cn) (accessed on 6 September 2023) [27,28].

### 5.10. Cell Culture and LPS/THz Treatment

HaCaT cell line was maintained in Dulbecco’s Modified Eagle’s Medium (DMEM) (Cytiva, Marlborough, MA, USA), containing 10% Fetal Bovine Serum (FBS) (BI, Israel) and 1% Penicillin–Streptomycin. For THz/LPS treatment, HaCaT cells were treated with lipopolysaccharide (LPS) (Sigma, Livonia, MI, USA) (5 μg/mL) for 12 h, during which time the cells were exposed to 0.1 THz irradiation twice, each for 10 min, with a 6 h gap between the two exposures.

### 5.11. RNA Extraction and RT-qPCR Program

Total RNA of HaCaT cells was extracted using a commercialized kit (Tiangen, Beijing, China) according to the manufacturer’s instructions. After reverse transcribing the RNA of the samples into cDNA, the primers for the target gene were designed to perform PCR amplification reaction. Following the reverse transcription of RNA from the samples into cDNA, RT-qPCR was performed to analyze the expression levels of the target genes with SYBR Green dye added to the reaction system. The primer sequences are listed in the Appendix A (online). The expression of genes was normalized to the geometric mean of the housekeeping gene GAPDH to control the variability in expression levels, and it was analyzed using the 2^−ΔΔCt^ method.

### 5.12. Western Blotting Analysis

Protein lysates of HaCaT cells were made using RIPA and PMSF (Beyotime, Shanghai, China). Protein samples were separated by SDS-polyacrylamide gel electrophoresis and transferred to a hydrophobic polyvinylidene fluoride (PVDF) membrane. After blocking with 5% BSA on a table concentrator for 90 min at room temperature, PVDF membranes were incubated with the primary antibody on a table concentrator at 4 °C overnight before being washed three times with TBST. After that, PVDF membranes were incubated with secondary antibodies on a table concentrator for 1 h at room temperature. The detailed information on antibodies can be found in Appendix A. The signals were detected with ECL (Beyotime, Shanghai, China) using a BIO-RAD ChemiDoc Touch Imaging system. Every experiment was repeated 3 times. Gray values analysis was quantified by the Image J software (National Institutes of Health, Bethesda, MD, USA).

### 5.13. Statistical Analysis

All the experimental data were shown using the mean ± standard error of the mean (SEM). The normality of all data was assessed using the Shapiro–Wilk test with GraphPad Prism 9 (GraphPad Software Inc., San Diego, CA, USA). Data with *p*-values > 0.05 were considered to be normally distributed and were subsequently analyzed through one-way ANOVA parametric analysis, followed by Tukey’s post hoc test. Specific symbols (*) indicate statistically significant differences as compared to the Control group for * *p* < 0.05, ** *p* < 0.01, and **** p* < 0.001. Specific symbols (^#^) indicate statistically significant differences as compared to the CIA group for ^#^ *p* < 0.05, ^##^ *p* < 0.01, and ^###^ *p* < 0.001.

## Figures and Tables

**Figure 1 ijms-25-05963-f001:**
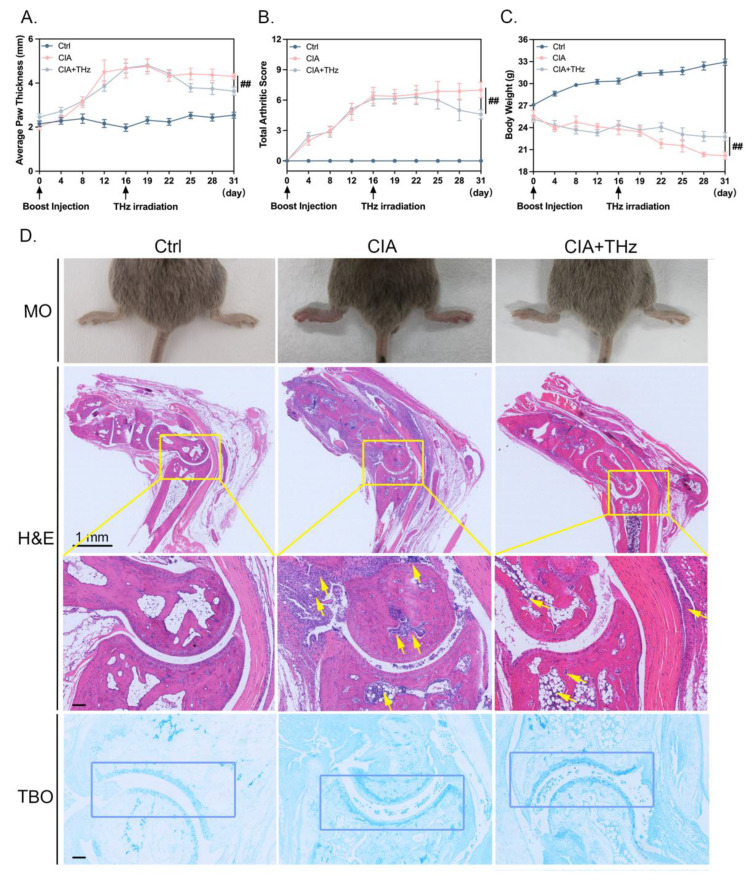
Irradiation of 0.1 sub-THz attenuated joint swelling in CIA mice. (**A**). Average paw thickness of mice (mm) from the day of boost injection. (**B**). Total arthritis score from the day of boost injection. (**C**). The body weight (g) from the day of boost injection. (**D**). Morphology (MO) of ankle joint, histopathological analysis (H&E), and Toluidine Blue O (TBO) staining of mice from Ctrl, CIA, and CIA + THz group, respectively. Data are presented as the mean ± SEM (*n* = 6 mice per group). Scale bar, 1 mm; scale bar in the enlarged diagram from the yellow box, 100 μm; the yellow arrows: neutrophil infiltration; the blue boxes: articular cartilage area. Symbol for the significance of differences compared to the CIA group: ^##^ *p* < 0.01.

**Figure 2 ijms-25-05963-f002:**
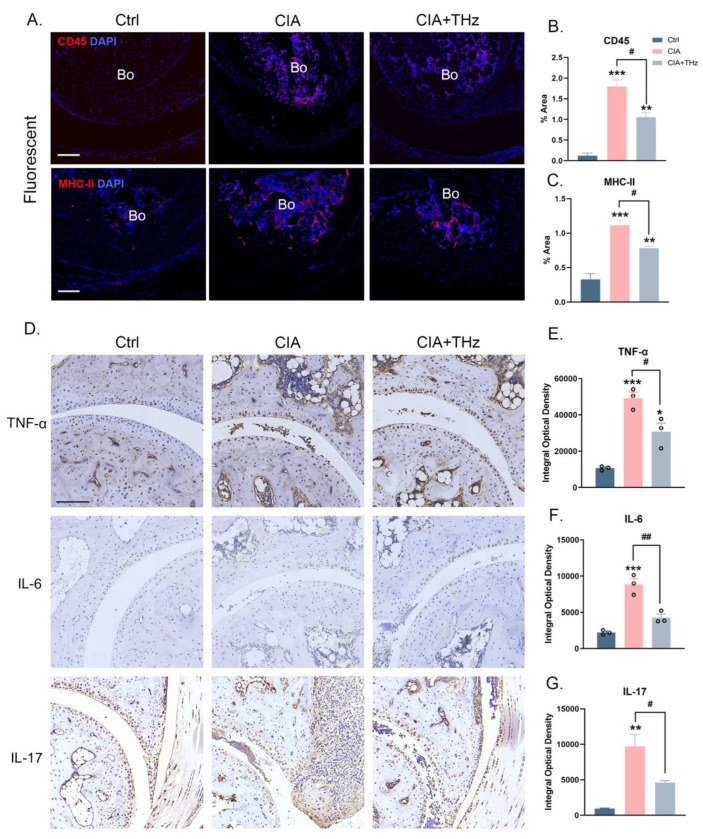
Irradiation of 0.1 sub-THz reduced infiltrated leukocytes and proinflammatory cytokine levels in the ankle joint. (**A**). IF staining of CD45 (red), MHC-II (red), and DAPI (blue) in ankle joint from Ctrl, CIA, and CIA+THz group, respectively. Scale bar, 100 μm. Bo: bone. (**B**). Percentages of positive area of inflorescence staining of CD45 (** *p* < 0.01, *** *p* < 0.001, and ^#^ *p* < 0.05). (**C**). Percentages of positive area of inflorescence staining of MHC-II (** *p* < 0.01, *** *p* < 0.001, and ^#^ *p* < 0.05). (**D**). IHC staining of TNF-α, IL-6, and IL-17 in ankle joint from each group, respectively. Scale bar, 100 μm. (**E**). Integral optical density of IHC staining of TNF-α (* *p* < 0.05, *** *p* < 0.01, and ^#^ *p* < 0.05). (**F**). Integral optical density of IHC staining of IL-6 (*** *p* < 0.001 and ^##^ *p* < 0.01). (**G**). Integral optical density of IHC staining of IL-17 (** *p* < 0.01 and ^#^ *p* < 0.05). Data are presented as the mean ± SEM (n = 3 mice per group). Symbol for the significance of differences compared to the Ctrl group: * *p* < 0.05, ** *p* < 0.01, *** *p* < 0.001; symbol for the significance of differences between the CIA group and CIA+THz group: ^#^ *p* < 0.05 and ^##^ *p* < 0.01.

**Figure 3 ijms-25-05963-f003:**
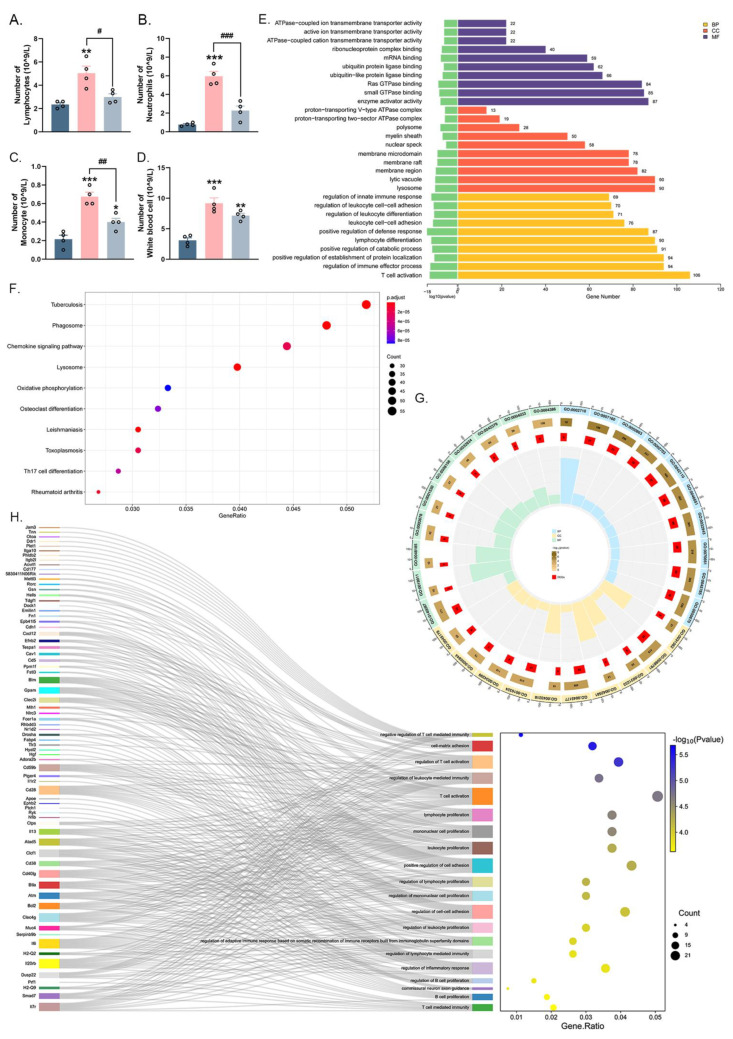
Analysis of blood cell components, murine hemocytes mRNA sequencing, and bioinformatics analysis. (**A**). Number of lymphocytes (** *p* < 0.01 and ^#^ *p* < 0.05). (**B**). Number of neutrophils (*** *p* < 0.001 and ^###^ *p* < 0.001). (**C**). Number of monocytes (* *p* < 0.05, *** *p* < 0.001, and ^##^ *p* < 0.01). (**D**). Number of white blood cells (** *p* < 0.05 and *** *p* < 0.001). (**E**). Top 10 GO analysis of DEGs from ANOVA analysis among Ctrl, CIA, and CIA+THz. (**F**). Top 10 KEGG pathway analysis of DEGs from ANOVA analysis among Ctrl, CIA, and CIA+THz. (**G**). Top 10 GO analysis of DEGs cut off from |log2Fold Change| > 0.67 and *p*-value < 0.05 between CIA and CIA+THz group. (**H**). Top 20 BP category in GO analysis of DEGs cut off from |log2Fold Change|>0.67 and *p*-value <0.05 between CIA and CIA+THz group. The GO analysis of DEGs including three main GO categories: Molecular Function (MF), Cellular Component (CC), and Biological Process (BP).

**Figure 4 ijms-25-05963-f004:**
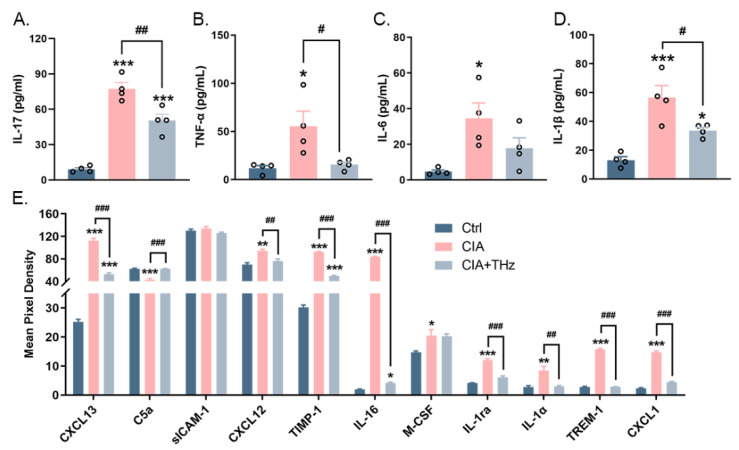
Detection of key cytokines’ content in serum. (**A**). Content of IL-17 in serum (*** *p* < 0.001 and ^##^ *p* < 0.01). (**B**). Content of TNF-α in serum (* *p* < 0.05 and ^#^ *p* < 0.05). (**C**). Content of IL-6 in serum (* *p* < 0.05). (**D**). Content of IL-1β in serum (* *p* < 0.05, *** *p* < 0.001 and ^#^ *p* < 0.05). (**E**). Mean pixel density of significant enriched cytokines. Data are presented as the mean ± SEM (n = 4 mice per group) (* *p* < 0.05, ** *p* < 0.01, and *** *p* < 0.001; ^#^ *p* < 0.05, ^##^ *p* < 0.01, and ^###^ *p* < 0.001). Symbol for the significance of differences compared to the Ctrl group: * *p* < 0.05, ** *p* < 0.01, and *** *p* < 0.001; symbol for the significance of differences between the CIA group and CIA+THz group: ^#^ *p* < 0.05, ^##^ *p* < 0.01, and ^###^ *p* < 0.001.

**Figure 5 ijms-25-05963-f005:**
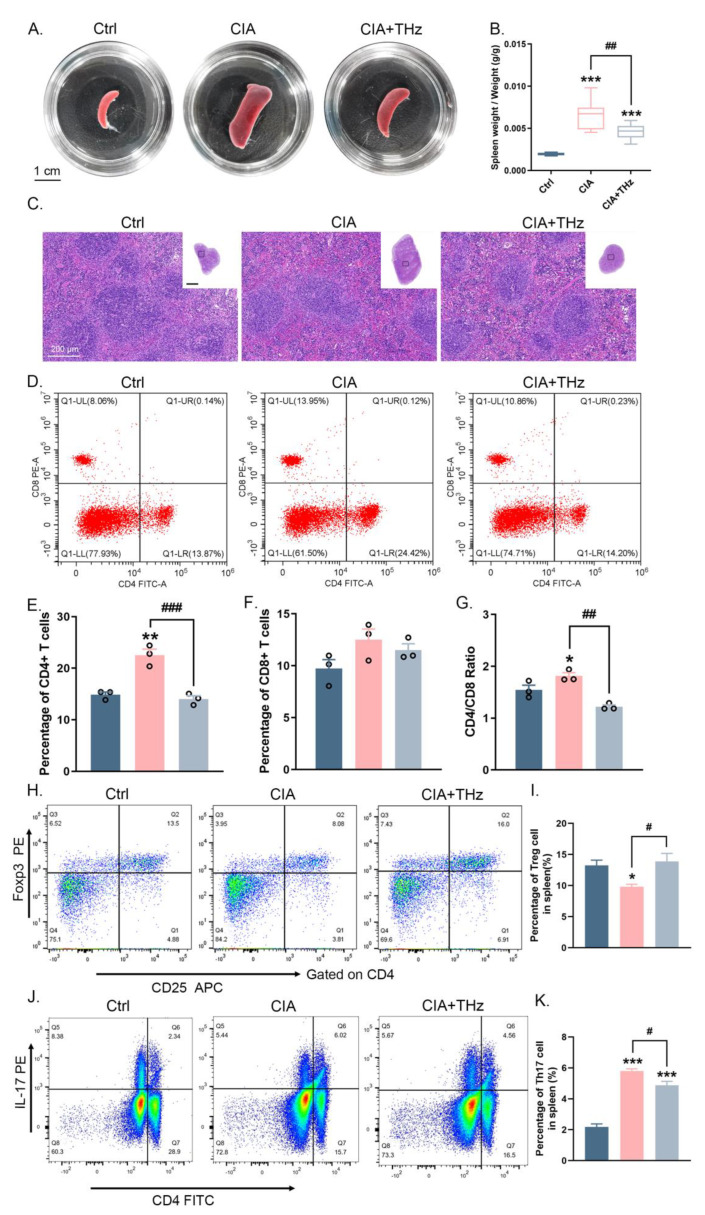
Radiation of 0.1 sub-THz reduced the immune activation in the spleen of CIA mice. (**A**). Morphology of mice spleen from Ctrl, CIA, and CIA+THz group, respectively. (**B**). Ratio of spleen weight and body weight of mice (*** *p* < 0.001 and ^##^ *p* < 0.01). (**C**). Overall vision (upper right corner) and local magnification of H&E staining of mice spleen from Ctrl, CIA, and CIA+THz group, respectively. Scale bar in overall vision, 2 mm; scale bar in local magnification, 200 μm; arrows denote macrophages. WhPu: white pulp; RePu: red pulp. (**D**). Dot plots of flow cytometry analysis indicating CD4+ and CD8+ T cell in splenic lymphocyte in each group. (**E**). Percentage of CD4+ T cell in splenic lymphocyte in each group (** *p* < 0.01 and ^###^ *p* < 0.001). (**F**). Percentage of CD8+ T cell in splenic lymphocyte in each group. (**G**). CD4 and CD8 ratio in each group (* *p* < 0.05 and ^##^ *p* < 0.01). (**H**). Dot plots of flow cytometry analysis indicating Treg cell in splenic lymphocyte in each group. (**I**). Percentage of Treg cell in splenic lymphocytes in each group (* *p* < 0.05 and ^#^ *p* < 0.05). (**J**). Dot plots of flow cytometry analysis indicating Treg cell in splenic lymphocyte in each group. (**K**). Percentage of Treg cell in splenic lymphocytes in each group. Data are presented as the mean ± SEM (n = 3 mice per group) (^#^ *p* < 0.05 and *** *p* < 0.001. Symbol for the significance of differences compared to the Ctrl group: * *p* < 0.05, ** *p* < 0.01, and *** *p* < 0.001; symbol for the significance of differences between the CIA group and CIA+THz group: ^#^ *p* < 0.05, ^##^ *p* < 0.01, and ^###^ *p* < 0.001.

**Figure 6 ijms-25-05963-f006:**
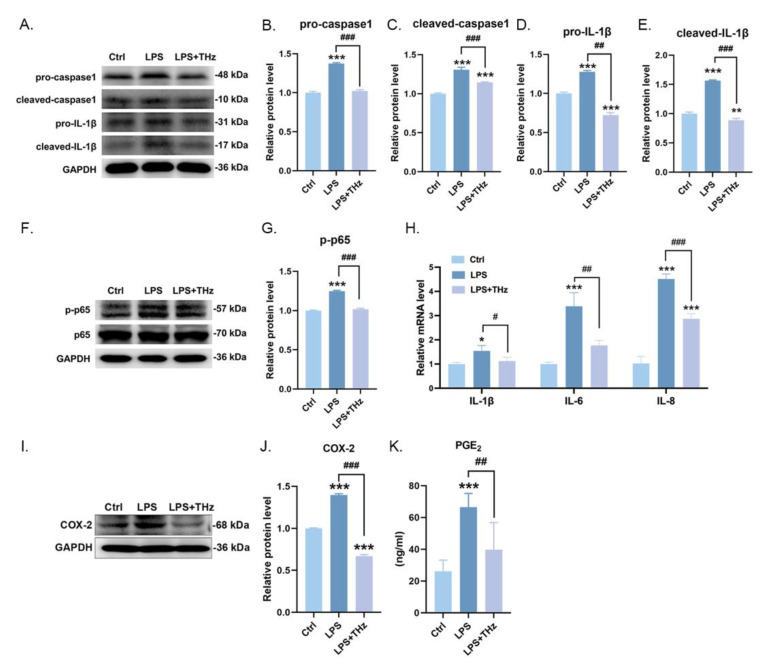
Irradiation of 0.1 sub-THz elevated the LPS-stimulated keratocytes inflammatory responses. (**A**). Western blotting of caspase 1, IL-1β, and GAPDH. (**B**). Relative protein level of pro-caspase 1 (*** *p* < 0.001 and ^###^ *p* < 0.001). (**C**). Relative protein level of cleaved-caspase 1 (*** *p* < 0.001 and ^###^ *p* < 0.001). (**D**). Relative protein level of pro-IL-1β (*** *p* < 0.001 and ^##^ *p* < 0.01). (**E**). Relative protein level of cleaved-IL-1β (** *p* < 0.01, *** *p* < 0.001, and ^###^ *p* < 0.001). (**F**). Western blotting of p65, p-p65, and GAPDH. (**G**). Relative protein level of p-p65 (*** *p* < 0.001 and ^###^ *p* < 0.001). (**H**). Relative mRNA level of IL-1β, IL-6, and IL-8 (* *p* < 0.05, *** *p* < 0.001, ^#^ *p* < 0.05, ^##^ *p* < 0.01, and ^###^ *p* < 0.001). (**I**). Western blotting of COX-2 and GAPDH. (**J**). Relative protein level of COX-2 (*** *p* < 0.001 and ^###^ *p* < 0.001). (**K**). PGE_2_ content in serum (* *p* < 0.05, and *** *p* < 0.001and ^#^ *p* < 0.05 and ^##^ *p* < 0.01). Data are presented as the mean ± SEM (n = 3 samples per group). Symbol for the significance of differences compared to the Ctrl group: * *p* < 0.05, ** *p* < 0.01, and *** *p* < 0.001; symbol for the significance of differences between the CIA group and CIA+THz group: ^#^ *p* < 0.05, ^##^ *p* < 0.01, and ^###^ *p* < 0.001.

## Data Availability

The mRNAseq datasets generated and/or analyzed during the current study are available in the GEO repository (GSE246029). (https://www.ncbi.nlm.nih.gov/geo/query/acc.cgi?acc=GSE246029). Other datasets used and/or analyzed during the current study are available from the corresponding author on reasonable request.

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
