# Peer review of "Anti-Inflammatory and Immunomodulatory Effects of 0.1 Sub-Terahertz Irradiation in Collagen-Induced Arthritis Mice"

_ijms, 2024, doi:10.3390/ijms25115963_

Round 1
Reviewer 1 Report
Comments and Suggestions for Authors
The authors assessed the anti-inflammatory and immunomodulatory effects of irradiation in collagen-induced arthritis model in mice. Overall, the study is well conducted, written in an organized, well understandable and coherent manner. I strongly recommend this manuscript for publication. A few points for authors' consideration are as follows:
1. Please improve the font size of the Figures 1A, 1B and 1C so that they are legible.
2. Similarly, improve the legibility of the Figure 3 for a better readability.
3. Please check the normality of the numerical data before employing one-way ANOVA test. Use either a Shapiro-Wilk or K-S test and then provide the results for every numerical variable that is analyzed and then state either a parametric or non-parametric test to be applied.
4. Please state limitations, strengths and future research directions before the conclusion section.
Reviewer 2 Report
Comments and Suggestions for Authors
This paper makes impressive use of biological markers, including cytokine levels, leukocyte enhancement, and altered the gene expression to show an effect of 0.1 THz radiation on live CIA mice and an immortalized keratinocyte cell line. If these effects were shown to be replicable with greater number of mice, the work would be of great importance.
There are, however, a number of issues that need some attention.
1) The references cited are for 2.5 THz, which is 25 times the energies of 0.1 THz radiation. This is of only marginal usefulness.
2) The needs to be a more comprehensive description of the incident radiation and its penetration:
i) Total area irradiated
ii) Power density delivered at various depths given the dielectric properties of the tissues. (the absorption coefficient of tissues at 0.1 THz is 60-80 cm-1, so the power density is reduced to 2% of its original incident value at 0.5 mm into the tissues).
3) It is always difficult to present the gene expression data in a form that is readable, but the fonts in fig. 3 are really unreadable, even at high magnification.
4) The Discussion needs to reflect some of the inherent limitations of the study, such as:
i) The low number of mice, (3 in each cohort).
ii) The issues raised with the penetration depth, and thus scalability of radiation to larger animals and humans.
iii) It may be better to limit the notes in the discussion to CIA, with a note that it is used as a mouse model of human RA.
Although difficult, at least some attempt at a mechanism would be useful.
Comments on the Quality of English Languageno issues,
